# Effects of Preoperative Anxiety on Postoperative Outcomes and Sleep Quality in Patients Undergoing Laparoscopic Gynecological Surgery

**DOI:** 10.3390/jcm12051835

**Published:** 2023-02-24

**Authors:** Xiangyi Gu, Yufei Zhang, Wenxin Wei, Junchao Zhu

**Affiliations:** Department of Anesthesiology, Shengjing Hospital of China Medical University, Shenyang 110055, China

**Keywords:** preoperative anxiety, sleep quality, pain, general anesthesia, satisfaction

## Abstract

Objective: Preoperative anxiety is a psychological state that commonly occurs before surgery and may have a negative impact on postoperative outcomes. This study aimed to investigate the effects of preoperative anxiety on postoperative sleep quality and recovery outcomes among patients undergoing laparoscopic gynecological surgery. Methods: The study was conducted as a prospective cohort study. A total of 330 patients were enrolled and underwent laparoscopic gynecological surgery. After assessing the patient’s preoperative anxiety score on the APAIS scale, 100 patients were classified into the preoperative anxiety (PA) group (preoperative anxiety score > 10) and 230 patients into the non-preoperative-anxiety (NPA) group (preoperative anxiety score ≤ 10). The Athens Insomnia Scale (AIS) was assessed on the night before surgery (Sleep Pre 1), the first night after surgery (Sleep POD 1), the second night after surgery (Sleep POD2), and the third night after surgery (Sleep POD 3). Postoperative pain was evaluated by the Visual Analog Scale (VAS), and the postoperative recovery outcomes and adverse effects were also recorded. Result: The AIS score in the PA group was higher than that of the NPA group at Sleep-pre 1, Sleep POD 1, Sleep POD 2, and Sleep POD 3 (*p* < 0.05). The VAS score was higher in the PA group than in the NPA group within 48 h postoperatively (*p* < 0.05). In the PA group, the total dosage of sufentanil was significantly higher, and more rescue analgesics were required. Patients with preoperative anxiety showed a higher incidence of nausea, vomiting, and dizziness than those without preoperative anxiety. However, there was no significant difference in the satisfaction rate between the two groups. Conclusion: The perioperative sleep quality of patients with preoperative anxiety is worse than that of patients without preoperative anxiety. Moreover, high preoperative anxiety is related to more severe postoperative pain and an increased requirement for analgesia.

## 1. Introduction

Sleep is a naturally occurring state of decreased arousal that is crucial for normal immune and cognitive function. Research in recent years has revealed that sleep function and sleep cycles may be altered perioperatively by surgery and other interventions under general anesthesia [1]. Poor sleep quality and postoperative insomnia not only lead to hyperalgesia and delayed recovery [2] but can increase the risk of potential complications, including cognitive impairment, chronic pain and emotional disturbances, metabolic disorders, and pro-inflammatory alteration [3,4]. General anesthesia is a state of hyporesponsiveness induced medically that resembles natural sleep. Studies have shown that general anesthesia can lead to decreases in rapid eye movement (REM) and slow-wave sleep (SWS), resulting in postoperative sleep disturbance [5,6]. Previous studies have also found that age, preoperative comorbidities, and severe surgical stimulation are independent risk factors associated with postoperative sleep disorders [7,8]. In addition, anxiety is an unpleasant sensation that compromises patients’ comfort and wellbeing. A study by Ruis et al., reported that anxiety commonly occurs in surgery patients, including fear of surgery and anesthesia-related fears, with an incidence of 25–80% [9]. Furthermore, high-anxiety states were investigated as a potential predictor of severe postoperative pain and postoperative complications such as increased postoperative morbidity and mortality [10,11]. Given that several prior studies have reported that preoperative anxiety has an effect on postoperative sleep quality in patients undergoing gynecological surgery, this study aimed to investigate the effects of preoperative anxiety on postoperative outcomes and sleep quality in patients undergoing gynecological surgery [12,13,14]. Studying these results could enable us to better manage patients during the perioperative period to promote their postoperative recovery.

## 2. Materials and Methods

The study was approved by the Human Research Ethics Committee of Shengjing Hospital, Shenyang, Liaoning Province, China (IRB registration number 2021PS664K), and it complied with the Declaration of Helsinki. Written informed consent was obtained from all patients participating in the trial. The trial was registered at Clinicaltrials.gov (first registration in 6 November 2020) before patient enrollment (NCT04619979).

### 2.1. Participants

This study enrolled patients undergoing gynecological surgery under general anesthesia at Shengjing Hospital of China Medical University (Figure 1). Patients aged 18–75 years with a grade I or II American Society of Anesthesiologists (ASA) score who underwent elective laparoscopic gynecological surgery lasting 1–3 h were enrolled in the study. The exclusion criteria were as follows: cardiovascular disease; chronic use of analgesics; chronic use of antidepressants; use of sleep-promoting drugs; sleep disorders; sleep apnea syndrome; history of abnormal surgery or recovery from anesthesia; psychosis; patients with impaired verbal communication; unwillingness to provide informed consent.

### 2.2. Sample Size

The prevalence of high preoperative anxiety was estimated to be 25% of the population according to previous reports [15,16]. The sample size was calculated using a formula for the estimation of a single population proportion: N = Z^2^_α/2_ p (1 − p)/d^2^ (N = minimum sample size; Z_α/2_ = standard normal variable value at 95% CI: 1.96; p = prevalence: 25%; d = margin of error: 5%). However, considering the non-response rate, and to avoid the underestimation of the prevalence of anxiety, the sample size was identified as 365, which is more than the formula result (N = 288).

### 2.3. Study Protocol

In this study, a laboratory assistant used the Amsterdam Preoperative Anxiety and Information Scale (APAIS) to assess preoperative anxiety in patients before premedication was administered [17,18]. The patients were divided into a preoperative anxiety group (PA group; preoperative anxiety score > 10) and a non-preoperative-anxiety group (NPA group; preoperative anxiety score ≤ 10), according to the APAIS total anxiety score. All patients fasted for 8–12 h before operation. Upon arriving in the operating room, electrocardiogram (ECG), peripheral oxygen saturation (SpO_2_), and non-invasive blood pressure (NIBP) were routinely monitored. The tracheal tube was inserted after induction of general anesthesia. The tidal volume was adjusted to 8 mL/kg by volume-controlled mechanical ventilation. The ventilation rate was then adjusted according to the end-tidal carbon dioxide (ETCO_2_) pressure to maintain ETCO_2_ at 35–45 mmHg. Propofol (4–10 mg/kg/h) and remifentanil (0.15–0.2 μg/kg/min) were used for intraoperative maintenance, while inhaling 50% oxygen (O_2_) fresh gas at a rate of 2 L/min. All anesthetic infusions were stopped 10 min before the end of the operation, and the patient was transferred to the PACU for continuous monitoring after extubation. Patient-controlled intravenous analgesia (PCIA) after the operation (sufentanil 2.0 μg/kg + normal saline 100 mL) was added to the intravenous analgesia pump. The infusion rate was 2 mL/h. The patient could press the pump according to the analgesic effect. The dosage of each pressing was 0.5 mL, and the locking time was 15 min.

### 2.4. Data Collection

The Athens Insomnia Scale (AIS) was assessed on the night before surgery (Sleep Pre 1) and on the first night (Sleep POD 1), the second night (Sleep POD 2), and the third night after surgery (Sleep POD 3) to evaluate subjective sleep quality. The AIS consists of 8 items: sleep induction, total sleep duration, nocturnal awakenings, final awakenings, sleep quality, functional capacity, wellbeing, and daytime sleepiness. The AIS is graded on a scale of 0–3, where “3” indicates a negative outcome. The total AIS score ranges between 0 and 24. A total score ≥ 6 points indicates a diagnosis of insomnia [19,20,21]. Pain intensity was assessed using the 10 cm Visual Analog Scale (VAS), where 0 cm indicates no pain and 10 cm indicates the most severe pain. VAS effectively controlled analgesia for 2, 4, 24, and 48 h postoperatively [22]. If the VAS score exceeded 6 cm, the patient was administered 50 mg of flurbiprofen axetil IV. The VAS for satisfaction (VAS-S) consists of a 10 cm long horizontal line with the descriptions “poor” on the far left (corresponding to 0 point) and “excellent” on the far right (corresponding to 10 point). The patient satisfaction scores are divided into “poor” (0~2 points), “moderate” (3~5 points), “good” (6~8 points), and “excellent” (9~10 points). Postoperative complications including bradycardia (heart rate below 50 beats per minute), hypotension (blood pressure below 20% of the basal value), respiratory depression (respiratory rate of fewer than 10 breaths per minute), dizziness (a reported sensation of unsteadiness accompanied by a feeling of movement within the head), and nausea and vomiting (any nausea, retching, or vomiting) during the first 24 h after surgery were recorded and treated accordingly.

## 3. Statistical Analyses

SPSS 23.0 (IBM Corp, Armonk, NY, USA) and GraphPad Prism8.0 statistical software were used for statistical data analysis. Normally distributed data are presented as the mean ± standard deviation ($\bar{x}$ ± s), and intergroup comparisons were conducted by using the *t*-test. Differences in AIS scores between the different time points were analyzed using the Greenhouse–Geisser test. Qualitative data are expressed as the number (n) and percentage (%) and were evaluated by the χ^2^ test; *p* < 0.05 (two-sided) was considered to represent a statistically significant difference.

## 4. Results

In the prospective observational study, 374 patients from Shengjing Hospital of China Medical University were included, of whom 9 patients were canceled scheduled surgeries. 365 patients were enrolled, of whom 9 patients were excluded according to the aforementioned criteria. Ultimately, 356 patients were included. The preoperative anxiety scores of the patients were evaluated on the APAIS. There were 110 patients in the preoperative anxiety group (preoperative anxiety score > 10), and 246 patients in the non-preoperative-anxiety group (preoperative anxiety score ≤ 10). Ten patients in the preoperative anxiety group and sixteen in the non-preoperative-anxiety group had incomplete data. Therefore, we finally analyzed data from 100 patients in the preoperative anxiety group and 230 patients in the non-preoperative-anxiety group (Figure 1).

### 4.1. Demographic Characteristics of the Two Groups

The two groups were similar in terms of patient characteristics, intraoperative parameters, and adverse events (Table 1). The preoperative anxiety scores of the PA group and the NPA group were 16.5 ± 4.1 and 4.4 ± 3.3, respectively (*p* < 0.001). Preoperative AIS scores showed no significant differences between the PA and NPA groups (*p* = 0.097). The difference in age between the PA group and the NPA group was statistically significant (*p* < 0.001). Elderly patients had lower APAIS scores in the preoperative period.

### 4.2. Comparison of Perioperative Sleep Quality between the PA Group and the NPA Group

The preoperative AIS scores and postoperative sleep quality of the two groups at different times are shown in Table 2 and Figure 2. There were significant differences in the AIS scores between the PA and NPA groups at different times after the operation (F = 114.275, *p* < 0.001). The AIS scores in both groups were highest on the first postoperative day and decreased within 3 days. Except for the fact that there was no significant difference in the AIS scores between the two groups before surgery, the AIS score of the PA group was higher at all other time points. We found no interaction between preoperative anxiety and each time point (F = 2.516, *p* = 0.114).

### 4.3. Differences in Postoperative Recovery Outcomes and Complications between the Two Groups

The VAS scores were higher in the PA group than in the NPA group at 2, 4, 24, and 48 h after the surgery (*p* = 0.012, *p* = 0.021, *p* = 0.028, and *p* = 0.002, respectively; Table 3). When the satisfaction rates of the patients were compared, the difference was statistically significant between the two groups (*p* = 0.023, Table 4). The incidences of dizziness and of nausea and vomiting were significantly higher in the PA group than in the NPA group (*p* = 0.038 and *p* = 0.045, respectively; Table 5). The length of hospital stay was also extended in the PA group compared with the NPA group (*p* = 0.046).

## 5. Discussion

The findings of our prospective observational study proved that patients are prone to experiencing preoperative anxiety before surgery. Patients with severe preoperative anxiety presented higher perioperative AIS scores and postoperative VAS scores than patients without anxiety. Moreover, preoperative anxiety patients had increased risks of suffering postoperative adverse effects such as nausea, vomiting, and dizziness.

Preoperative anxiety, defined as “an unpleasant restless or tense state in which the patient is worried about illness, hospitalization, anesthesia, and surgery or the unknown”, is an important problem encountered by patients, as it can lead to emotional, mental, and physical problems [23]. With more than 312.9 million surgeries performed worldwide each year, there is an urgent need to better assess individual patient perceptions of procedures and outcomes. An estimated 25–80% of patients will experience anxiety before surgery, which can cause hyperalgesia and delayed recovery [9,24,25]. Perioperative anxiety was associated with increased autonomic nervous fluctuations and increased demand for anesthesia, increased incidence of nausea and vomiting, and increased postoperative pain [26]. According to reports, these complications can prolong both the recovery period and the hospital stay. In our study, many patients had high levels of anxiety before surgery, and all patients had varying degrees of anxiety. Anxiety may develop due to fear of anesthesia, surgery, and several other different reasons [27,28]. This prospective observational study found that 30% of patients presented preoperative anxiety, with an APAIS score > 10, almost half of whom presented with a high degree of anxiety (APAIS score > 20). By investigating the perioperative sleep quality of the enrolled patients, we found that the AIS scores were higher in the PA group than in the NPA group on the night before surgery. We observed a significant difference in the AIS scores between the two groups compared to the previous time point, which is consistent with the findings of a previous study [29]. Furthermore, the PA group also had significantly higher AIS scores than patients in the NPA group for the first 3 days after surgery. According to the biological anxiety theory, the autonomic nervous system of anxious individuals adapts slowly to repetitive stimuli, while overreacting to moderate stimuli. The reason for the observed dramatic changes in intraoperative heart rate, contractility, and peripheral vascular resistance involves the activation of the hypothalamic–pituitary–adrenal axis due to increased cortisol secretion [30]. Previous studies reported and suggested that anxiety may cause greater intraoperative hemodynamic changes. Furthermore, for anesthesiologists, one of the most important consequences of primary anxiety is pain [31]. Ip et al., previously observed that preoperative anxiety significantly affected postoperative pain severity [32]. A meta-analysis from this study revealed a significant positive correlation between preoperative anxiety and the severity of postoperative acute pain [33,34]. Anxiety in the context of “catastrophic pain”, i.e., a strong negative reaction to real or expected pain, is also linked to severe postoperative pain [35]. The higher the degree of preoperative anxiety, the higher the requirement for postoperative analgesia [36]. Anesthesiologists and surgeons should therefore pay more attention to patients’ preoperative adverse emotions to allow a timely adjustment of perioperative medication to achieve individualized management. Our study found that the VAS scores were significantly higher in the PA group than in the NPA group at 2, 4, 24, and 48 h after the surgery. Furthermore, patients with preoperative anxiety developed more postoperative complications, including nausea, vomiting, and dizziness. Lower satisfaction rates were observed in the PA group. We found there was no significant difference between the two groups in terms of education levels (*p* = 0.713). However, increased anxiety scores were reported in those with high education levels by Cauomo et al. [37]. It was proposed that different assessment tools and analytical methods can lead to differences in conclusions.

Whether short-term perioperative psychological counseling can effectively improve preoperative anxiety still remains to be studied in patients with preoperative anxiety. This study has several limitations that should be noted. First, this study involved a single-center survey, which reduces the generalizability of the results. Second, we only followed up the short-term postoperative data, meaning that long-term prognostic indicators need to be further explored. Third, previous studies have shown that female patients are more prone to preoperative anxiety than male patients [38]; however, in this study, we only observed and compared female patients undergoing gynecological surgery. Other patient groups, such as elderly patients and pediatric patients, should be investigated in future work.

## 6. Conclusions

Our findings show that the perioperative sleep quality of patients undergoing laparoscopic gynecological surgery with preoperative anxiety is worse than that of patients without preoperative anxiety. Moreover, high preoperative anxiety is related to more severe postoperative pain and an increased requirement for analgesia. However, large-scale multicenter trials are needed to confirm these findings and provide early identification and intervention strategies to improve the sleep quality of postoperative patients.

## Figures and Tables

**Figure 1 jcm-12-01835-f001:**
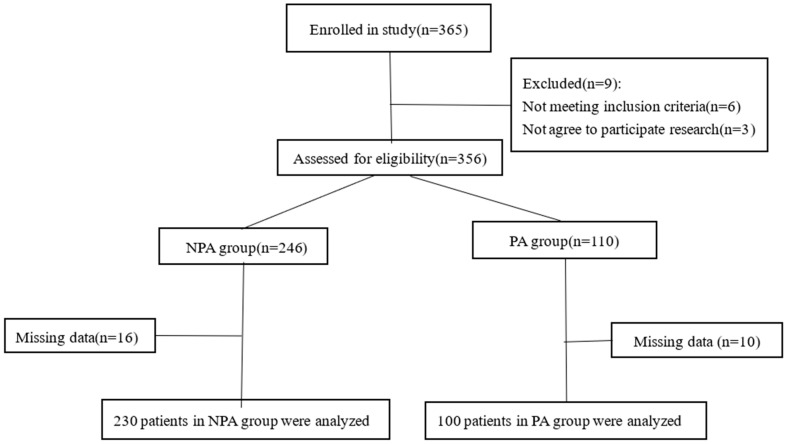
Study flowchart.

**Figure 2 jcm-12-01835-f002:**
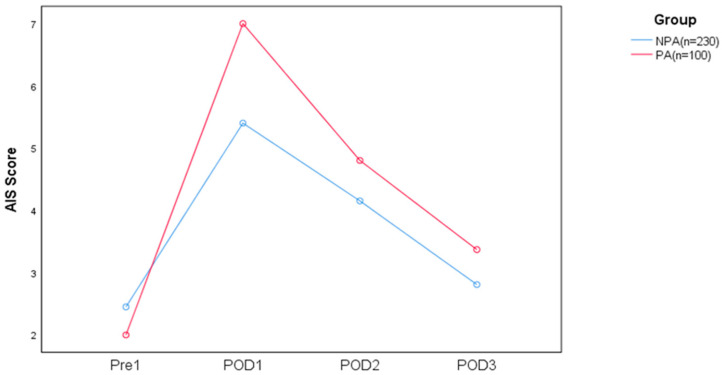
Comparison of sleep quality between the PA and NPA groups.

**Table 1 jcm-12-01835-t001:** Demographic information and intraoperative parameters.

	NPA Group n = 230	PA Group n = 100	*P* Value
Age n (%)			<0.001
<35	44 (19.1%)	40 (40.0%)	
35–55	148 (64.3%)	53 (53.0%)	
>55	38 (16.6%)	7 (7%)	
BMI (Kg/m^2^)	23.9 ± 3.9	24.6 ± 5.0	0.441
Preoperative APAIS	4.4 ± 3.3	16.5 ± 4.1	<0.001
Preoperative AIS score	2.4 ± 2.2	2.0 ± 2.3	0.097
Education level n (%)			0.713
Middle school	25 (10.8%)	14 (14.0%)	
High school	134 (58.3%)	57 (57.0%)	
University and above	71 (30.9%)	29 (29.0%)	
History of motion sickness	0 (0)	0 (0)	-
Comorbidities n (%)			0.669
Cardiovascular disease	1 (0.4%)	1 (1.0%)	
Hypertension	8 (3.5%)	2 (2.0%)	
Diabetes	5 (2.2%)	2 (2.0%)	
Reasons for surgery n (%)			0.739
Endometrial intraepithelial neoplasia	39 (17.0%)	12 (12.0%)	
Ovarian or fallopian tube cysts	98 (42.6%)	49 (49.0%)	
Uterine fibroids	76 (33.0%)	31 (31.0%)	
Endometriosis	3 (1.3%)	1 (1.0%)	
Adenomyosis	14 (6.1%)	7 (7.0%)	
Surgical methods n(%)			0.289
Hysterectomy	42 (18.3%)	9 (9.0%)	
Ophorectomy	45 (20.0%)	17 (17.0%)	
Tubectomy	13 (5.7%)	6 (6.0%)	
Salpingo-oophorectomy	43 (18.7%)	23 (23.0%)	
Myomectomy	72 (31.3%)	35 (35.0%)	
Excision of uterine lesions	15(6.5%)	10 (10.0%)	
Duration of surgery (h)	1.5 ± 1.0	1.6 ± 0.8	0.405
Duration of anesthesia (h)	1.8 ± 0.9	1.9 ± 1.0	0.329
Remifentanil consumption (μg/kg)	6.5 ± 3.8	6.6 ± 4.2	0.855
Hypotension, n (%)	36 (16.7%)	12 (12.0%)	0.414
Respiratory depression, n (%)	0 (0)	0 (0)	-
Bradycardia, n (%)	8 (3.5%)	3 (3.0%)	0.841

Data are presented as the mean (±SD) or as n (%).

**Table 2 jcm-12-01835-t002:** Impacts of preoperative anxiety on postoperative sleep quality.

Group	AIS Score				Sum	F	*P* Value
	Pre1	POD1	POD2	POD3			
NPA	2.4 ± 2.2	5.4 ± 2.6	4.2 ± 2.1	2.8 ± 1.6	3.7 ± 2.1		
PA	2.0 ± 2.3	7.0 ± 2.5	4.8 ± 2.4	3.4 ± 1.7	4.3 ± 2.2		
Sum	2.2 ± 2.3	6.2 ± 2.5	4.5 ± 2.3	3.1 ± 1.7	4.0 ± 2.2 *	114.275 *	<0.001
*t*	1.664	5.276	2.434	2.918	14.529 *	F = 2.516, *P* = 0.114 ^#^	
*P*	0.097	<0.001	0.015	0.004	<0.001		

* F statistic and *P* value of main effect; ^#^ F statistic and *P* value of crossover effect.

**Table 3 jcm-12-01835-t003:** Postoperative pain.

	NPA Group n = 230	PA Group n = 100	*P* Value
VAS 2 h	3.9 ± 1.6	4.5 ± 1.5	0.012
VAS 4 h	3.3 ± 1.7	3.6 ± 1.3	0.021
VAS 24 h	2.2 ± 1.5	2.5 ± 1.4	0.028
VAS 48 h	1.4 ± 1.3	1.5 ± 1.2	0.002
Total Sufentanil consumption (mg)	76.9 ± 37.8	88.4 ± 35.4	0.008
Flurbiprofen axetil requirement n (%)	11 (4.8%)	6 (6.0%)	0.036

Data are presented as mean (±SD) or as n (%).

**Table 4 jcm-12-01835-t004:** Postoperative Patient Satisfaction.

	NPA Group n = 230	PA Group n = 100	*P* Value
			0.023
Bad	0 (0)	0 (0)	
Mild	58 (25.2%)	34 (34.0%)	
Good	84 (36.5%)	43 (43.0%)	
Excellent	88 (38.3%)	23 (23.0%)	

Data are presented as n (%).

**Table 5 jcm-12-01835-t005:** Postoperative recovery and complications.

	NPA Group n = 230	PA Group n = 100	*P* Value
Nausea and vomiting, n (%)	15 (4.5%)	59 (17.9%)	0.038
Dizzy, n (%)	4 (1.2%)	13 (3.9%)	0.045
Hospital stay (day)	9	10	0.046
Other		1 (delirium)	

Data are presented as n (%).

## Data Availability

The datasets used and/or analyzed during the current study are public data available from the corresponding author upon request (zhujunchao1@hotmail.com).

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
