# Peer review of "Effects of Preoperative Anxiety on Postoperative Outcomes and Sleep Quality in Patients Undergoing Laparoscopic Gynecological Surgery"

_jcm, 2023, doi:10.3390/jcm12051835_

Round 1
Reviewer 1 Report
Interesting argument, but you choose a specific surgery where the patients are only females! Its known that there are differences between the sex about the anxiety…….and also the age.
Why you choose APAIS Scale ?
I’m not sure that is correct speak about PICA and not PCA, please check
References about anxiety in gynecological surgery is only one: Number 31, but you in your article speak about many references…….
Reference 24: Effect and not Efect
All references need to be standardized
Reviewer 2 Report
In this manuscript, the authors “aimed to investigate the effect of preoperative anxiety on postoperative sleep quality and recovery outcomes among patients undergoing gynecological surgery.” For that they conducted a prospective cohort study enrolling patients who underwent laparoscopic hysterectomy. The patient's preoperative anxiety was evaluated using APAIS scale and assessed the Athens Insomnia Scale on the night before surgery), on the first night after surgery, on the second night (Sleep POD2), and on the third night after surgery. They concluded, "The perioperative sleep quality of patients with preoperative anxiety is worse than that of patients without preoperative anxiety.”
The manuscript is well written and produced according to the reporting standards; the subject may be relevant and important, and the information provided is clear and objective.
Major issues
- Sample size calculation is a very important aspect of any study. It should be done at the time of planning a study, based on the type of research question and study design. There is no reference to the sample size calculation.
- We only are told that enrolled patients were ASA Physical status I and II but comorbidities are not described nor any aspect evaluation of risk (for instance ASA Physical status or Charlson Comorbidity Index). Were cancer patients excluded because they were considered all ASA III or more? That is an important aspect because without this knowledge authors cannot state “patient characteristics” are similar. We know very few variables about patient characteristics (age, education level, and BMI).
- Hysterectomy is a common gynecological surgery performed to remove the uterus in women with uterine myoma, endometriosis, uterine prolapse, genital cancers, and other benign conditions. The reason to surgery was not subjected to description and may influence preoperative anxiety. Is it possible to have data on reasons for surgery?
- Women who underwent a hysterectomy may have a risk of psychiatric morbidity and Women with depression were reported to fare worse after undergoing a hysterectomy. This may be considered another potentially confounding for this study. The study of depression at least should have been known.
- Postoperative complications including bradycardia, hypotension, respiratory depression, dizziness, nausea and vomiting during the first 24 h after surgery were recorded and treated accordingly. The definition for those complications was not described.
- Patient-controlled intravenous analgesia (PICA) after the operation was delivered to every patient but it is not described for how long. Only if VAS score exceeded 8 cm, the patient had a rescue analgesic. It is a very high perceived pain before any analgesia was administered.
- The patient satisfaction score was divided into “poor,” “moderate,” “good,” and “excellent.”. How was that measured, with what scale or questionnaire, and when and by whom?
- Study conclusions are overstated: I think the correct conclusion is that“ The perioperative sleep quality of patients with preoperative anxiety is worse than that of patients without preoperative anxiety.” But after that “Overall, this suggests that preoperative anxiety and its adverse effect are often underestimated.” Was not a study subject and the authors do not studied if data are underestimated or not. The same for “Anesthesiologists and surgeons should therefore pay more attention to patients’ preoperative adverse emotions to allow a timely adjustment of perioperative medication to achieve individualized management.” These sentences may be added to discussion section and should not be considered conclusions.
Round 2
Reviewer 2 Report
I have no further comments on the manuscript. The revised manuscript has acceptable answers to the questions posed by me as a reviewer in the first revision made by the authors. The manuscript is clearly improved and I sustain that the proposed subject is relevant and important and the paper in general is of interest. That justifies my decision to recommend the editorial team accept the manuscript's publication.
Author Response
We thank the reviewer for his comment and great help for improvement of our manuscript.